# Cytosolic Hsp70 and co-chaperones constitute a novel system for tRNA import into the nucleus

**Akira Takano[1], Takuya Kajita[1], Makoto Mochizuki[1], Toshiya Endo[1,2], Tohru Yoshihisa[3]***

[1]Department of Chemistry, Graduate School of Science, Nagoya University, Nagoya, Japan; [2]Faculty of Life Sciences, Kyoto Sangyo University, Kyoto, Japan; [3]Graduate School of Life Science, University of Hyogo, Kobe, Japan

**Abstract** tRNAs are unique among various RNAs in that they shuttle between the nucleus and the cytoplasm, and their localization is regulated by nutrient conditions. Although nuclear export of tRNAs has been well documented, the import machinery is poorly understood. Here, we identified Ssa2p, a major cytoplasmic Hsp70 in *Saccharomyces cerevisiae*, as a tRNA-binding protein whose deletion compromises nuclear accumulation of tRNAs upon nutrient starvation. Ssa2p recognizes several structural features of tRNAs through its nucleotide-binding domain, but prefers loosely-folded tRNAs, suggesting that Ssa2p has a chaperone-like activity for RNAs. Ssa2p also binds Nup116, one of the yeast nucleoporins. Sis1p and Ydj1p, cytoplasmic co-chaperones for Ssa proteins, were also found to contribute to the tRNA import. These results unveil a novel function of the Ssa2p system as a tRNA carrier for nuclear import by a novel mode of substrate recognition. Such Ssa2p-mediated tRNA import likely contributes to quality control of cytosolic tRNAs.

***For correspondence:** tyoshihi@sci.u-hyogo.ac.jp

**Competing interests:** The authors declare that no competing interests exist.

## Introduction

Most cytoplasmic RNAs are exported unidirectionally from the nucleus across the nuclear envelope (NE) after their birth and appropriate processing in the nucleus (*Lei and Silver, 2002*; *Grünwald et al., 2011*). tRNAs, however, are unique among major classes of cytoplasmic RNAs in that they shuttle between the nucleus and the cytoplasm (*Takano et al., 2005*; *Shaheen and Hopper, 2005*; for review; *Yoshihisa, 2006*; *Phizicky and Hopper, 2010*; *Hopper, 2013*). The first indication of nuclear-cytoplasmic shuttling of tRNAs was the discovery of cytoplasmic splicing of pre-tRNAs in the yeast *Saccharomyces cerevisiae* (*Yoshihisa et al., 2003*, *2007*) in spite of the existence of small amounts of mature tRNAs in the nucleus (*Sarkar and Hopper, 1998*; *Grosshans et al., 2000*). Subsequently, we and others demonstrated that mature tRNAs move from the cytoplasm back into the nucleus (*Shaheen and Hopper, 2005*; *Takano et al., 2005*). Nuclear import of tRNAs has also been observed in mammals (*Zaitseva et al., 2006*; *Shaheen et al., 2007*; *Miyagawa et al., 2012*), suggesting that eukaryotic cells are equipped with mechanisms that allow bidirectional movement of tRNAs between the nucleus and the cytoplasm.

The export of tRNAs has been well studied. Importin β-family proteins have been identified as tRNA carriers across the nuclear pore complex (NPC) on the NE. Los1p/exportin-t is considered as a primary export carrier of mature tRNAs in yeast, plants, and vertebrates (*Arts et al., 1998*; *Hellmuth et al., 1998*; *Kutay et al., 1998*; *Sarkar and Hopper, 1998*; *Li and Chen, 2003*; *Park et al., 2005*; *Cook et al., 2009*). Besides, yeast Los1p exports pre-tRNAs for cytoplasmic splicing (*Sarkar and Hopper, 1998*; *Yoshihisa et al., 2003*). Msn5p/exportin-5, another importin β homologue, provides an alternative export pathway in yeast and mammals (*Bohnsack et al., 2002*; *Calado et al., 2002*;

**eLife digest** Plants, animals, and fungi all store their DNA inside their cells within a structure called the nucleus, which is surrounded by a nuclear envelope that separates it from the rest of the cell. This DNA contains the instructions to build proteins, but proteins are actually built elsewhere in the cell, outside of the nucleus. This means that the instructions must first be copied and then carried out through pores in the nuclear envelope before they can be decoded to build the protein.

Outside of the nucleus, molecules called transfer RNAs (or tRNAs for short) are involved in the decoding process by carrying the required building blocks to the cell's protein-making machinery. The tRNA molecules can also shuttle back and forth, in and out of the nucleus. In yeast and mammals, the localization of the tRNA molecules depends on the availability of nutrients; if nutrients are scarce, then more tRNAs are moved into the nucleus. While the mechanism by which tRNAs exit the nucleus is well characterized, little is known about the movement in the opposite direction.

Takano et al. have now analyzed how tRNAs are transported into the nucleus in yeast cells by identifying the proteins that bind tightly to them. These experiments revealed that a protein called Ssa2p interacts with tRNAs. This protein belongs to a large family of proteins called Hsp70 chaperones that assist other proteins to fold into their correct shapes.

Takano et al. observed that Ssa2p tends to bind to tRNAs that are poorly folded, suggesting that it may work like other well-known chaperones. Furthermore, cells that lack Ssa2p were shown to be unable to efficiently transport tRNAs into the nucleus when nutrients were limited. These results together demonstrate that the yeast protein Ssa2p acts as a carrier for transporting tRNAs into the nucleus; further work is now required to understand whether this mechanism is also found in other species.

*Takano et al., 2005*; *Okada et al., 2009*) while Msn5p/exportin-5 is the main carrier of tRNAs in *Drosophila* (*Büssing et al., 2010*). These importin β proteins appear to play different but overlapping roles in the export of mature tRNAs in the yeast; Los1p exports both newly synthesized and re-imported tRNAs while Msn5p exports only re-imported tRNAs (*Eswara et al., 2009*; *Murthi et al., 2010*). In both cases, the export of spliced or mature tRNAs is coupled to a tRNA quality control step governed by nuclear aminoacyl-tRNA synthetases (ARSs) to support efficient export of aminoacylated tRNAs (*Lund and Dahlberg, 1998*; *Azad et al., 2001*). Efficient tRNA export relies on both nuclear and cytoplasmic factors, like Utp8p, Utp9p, Cex1p, so on. (*McGuire and Mangroo, 2007*; *Strub et al., 2007*; *Eswara et al., 2009*; *Nozawa et al., 2013*).

In contrast, mechanisms of tRNA import into the nucleus are only poorly characterized. Nuclear import of tRNAs can be blocked by depletion of intracellular ATP, indicating that the process is energy-dependent. However, tRNA import can also operate even in the absence of a gradient of GTP-bound Ran across the NE, which is the main energy source for many macromolecular transport mediated by importin β (*Takano et al., 2005*; *Grünwald et al., 2011*). It is known that various types of tRNAs are imported into the nucleus. Both authentic tRNAs and some types of damaged tRNAs, such as CCA-less tRNAs, are imported (*Takano et al., 2005*). Wybutosine formation on tRNA-Phe$_{GAA}$ is initiated by nuclear Trm5p after cytoplasmic splicing of pre-tRNA-Phe$_{GAA}$, indicating that nuclear import of a spliced intermediate precedes this modification (*Ohira and Suzuki, 2011*). In addition, spliced tRNA species but hypomodified and possessing 5′- and 3′ extensions, which are accidentally leaked from the nucleus in certain mutants, are retrograded into the nucleus to be repaired or degraded (*Kramer and Hopper, 2013*). These findings are in great contrast to the above-mentioned selective export of aminoacylated mature tRNAs. Collectively, tRNA import and export mechanisms with different specificities were predicted to maintain the quality of pre-existing mature tRNAs in the cytoplasm (*Yoshihisa, 2006*; *Kramer and Hopper, 2013*). However, identity of the carrier that mediates tRNA import remains totally unknown. One possible import carrier is an importin-β homologue Mtr10p, which was originally identified as an import carrier for Npl3p, as mRNA-binding protein (*Senger et al., 1998*), and was shown to cause a defect in tRNA import when mutated (*Shaheen and Hopper, 2005*). Nevertheless, no evidence has been reported for tRNA binding by Mtr10p.

Notably, balance between the import and export of tRNAs across the NE is determined by the physiological conditions. For instance, depletion of various nutrients, such as amino acids,

phosphate, and glucose, results in nuclear accumulation of tRNAs in *S. cerevisiae* (*Hurto et al., 2007*; *Whitney et al., 2007*). Indeed, such alteration of tRNA localization affects translation efficiency of certain mRNAs that encode enzymes for amino acid biosynthesis, suggesting that retrieval of tRNAs from the cytosol has, at least, a regulatory role (*Chu and Hopper, 2013*). To alter the tRNA balance, signal transduction pathways mediated by PKA and TOR, but not Gcn2p, facilitate this transport regulation in the yeast (*Whitney et al., 2007*). Although several kinases were shown to act on the regulation of tRNA transport, tRNA export carriers Los1p or Msn5p are not phosphorylated in vivo according to the nutrient status, and they are not substrates for PKA in vitro (*Pierce et al., 2014*). Similar regulation could operate not only in fungi, but also in mammals (*Shaheen et al., 2007*; *Miyagawa et al., 2012*), while some reports argue that this is specific to some fungal species (*Chafe et al., 2011*). A model was proposed that, while tRNA import is constitutive, export is fine-tuned, depending on the growth conditions (*Eswara et al., 2009*; *Murthi et al., 2010*). However, evidence is lacking for constitutive import of tRNAs at the same rate irrespective of nutrient conditions. It is thus essential to identify the factors mediating the tRNA import, which are possible targets of the regulation.

Here, we describe the identification and characterization of Ssa2p, one of the major cytosolic Hsp70 in budding yeast (*Werner-Washburne et al., 1987*), as a potential tRNA import carrier. While Hsp70 proteins are a class of molecular chaperones that bind and release proteins with exposed hydrophobic segments in an ATPase cycle-dependent manner to affect protein conformation (*Young et al., 2004*; *Kampinga and Craig, 2010*), in vivo and in vitro data indicate that Ssa2p plays a pivotal role in nuclear import of tRNAs, and that this process is achieved through a novel mode of substrate recognition through its nuclear binding domain (NBD).

## Results

### Identification of Ssa2p as a novel tRNA import factor

To search for carrier(s) of tRNA import into the nucleus, a biochemical approach was adopted to identify new tRNA-binding proteins. We previously showed that both full-length and CCA-less tRNAs are imported into the nucleus, and that this nuclear import is ATP-dependent (*Takano et al., 2005*). Thus, we postulated that the putative import carrier(s) does not recognize the 3′ end of tRNAs and binds tRNAs in a nucleotide triphosphate-dependent or sensitive manner. To identify such proteins, tRNA-agarose was prepared, in which yeast tRNAs were immobilized via their 3′ ends (*Figure 1A*). Next, a yeast cytosolic fraction depleted of endogenous tRNAs by anion exchange chromatography was applied to the tRNA-agarose in the presence or absence of 3 mM ATP. Bound proteins were subsequently eluted from the tRNA-agarose with 1.5 M NaCl. As shown in *Figure 1B*, the intensity of some bands varied depending on the absence (*Figure 1B* closed arrowheads) or presence of ATP (open arrowheads). Bands in the eluates marked by arrows were then subjected to peptide fingerprinting. As summarized in *Figure 1C*, Tef1p (eEF1A in the yeast) and Eno2p (enolase) were identified, both of which are already known as tRNA binding proteins (*Nagata et al., 1984*; *Entelis et al., 2006*). In addition, RNA binding proteins, such as Pab1p, Gbp2p and Sro9p, were also detected. Among the proteins associated with the tRNA-resin in an ATP-sensitive manner, Ssa1p and/or Ssa2p, two major cytosolic Hsp70s with highly homologous sequences, were of interest because mammalian Hsp70 (Hsc70) was shown to interact with an AU-rich element, which defines short-lived mRNAs (*Henics et al., 1999*; *Laroia et al., 1999*; *Lu et al., 2006*). In addition, the yeast *ssa1* mutant strain exhibits defective degradation of mRNAs possessing this AU-rich element (*Duttagupta et al., 2003*). Therefore, the Ssa proteins were selected for further analyses.

Because the peptide mass fingerprint of the tryptic fragments from the 70 kDa band could not discriminate between Ssa1p and Ssa2p (*Figure 1—figure supplement 1*), the *SSA1* or *SSA2* gene on the yeast chromosome was replaced with a FLAG-tagged version, and the capability of each FLAG-tagged Ssa protein to bind tRNAs in vivo was examined by RNA immunoprecipitation. When analyzed by Western blotting with the anti-FLAG antibody and by Northern blotting with an anti-mature tRNA-Pro$_{UGG}$ probe (*Figure 1D*), only immunoprecipitates of Ssa2p-FLAG in the absence of ATP contained tRNA-Pro$_{UGG}$ at levels above background, indicating that the Ssa protein we identified as a tRNA-binding protein was Ssa2p.

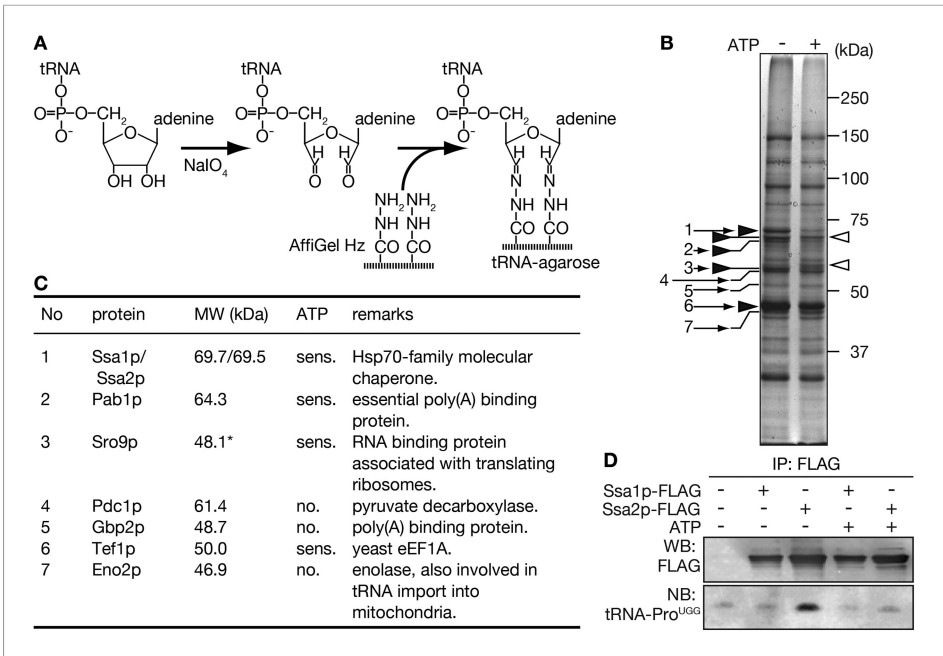

**Figure 1**. Purification of tRNA-interacting proteins with immobilized tRNA-resin. (**A**) A schematic diagram for preparation of tRNA-agarose by hydrazide coupling. (**B**) tRNA-binding proteins purified with the tRNA-agarose were analyzed by SDS-PAGE/CBB staining. Purification was performed in the absence (–) or presence (+) of 3 mM Mg-ATP. Bands appearing mainly in the ATP minus or plus lanes are marked by closed and open arrowheads, respectively. Bands that were identified by peptide mass fingerprinting after in-gel digestion are indicated by small numbered arrows. (**C**) A summary of the proteins identified in **B**. The numbers correspond to the arrows shown in **B**. *Sro9p, with a calculated molecular mass of 48.1 kDa, is known to migrate as a 60-kDa band on SDS-PAGE (*Sobel and Wolin, 1999*). (**D**) Yeast lysates prepared from strains expressing either Ssa1p-FLAG or Ssa2p-FLAG in addition to the wild type-strain were subjected to immunoprecipitation with anti-FLAG agarose in the absence (–) or presence (+) of 3 mM Mg-ATP. One-tenth of the eluates were analyzed by Western blotting with the anti-FLAG antibody (WB), and the remainders of the eluates were subjected to RNA extraction and Northern blotting with a probe against mature tRNA-Pro_{UGG} (NB).

The following figure supplement is available for figure 1:

**Figure supplement 1**. Identification of Ssa1p/Ssa2p as tRNA-binding protein by peptide mass fingerprinting.

Next, the question whether Ssa2p is indeed involved in the nuclear import of tRNAs in vivo was addressed. tRNA import was assessed by measuring nuclear accumulation of tRNAs in a *los∆1 msn5∆* double mutant (*Takano et al., 2005*) and by examining nuclear accumulation of tRNAs under nutrient starvation conditions (*Shaheen and Hopper, 2005*). First, an *ssa1∆* or *ssa2∆* mutation was introduced into the *los1∆ msn5∆* strain, and the localization of tRNA-Pro_{UGG}, encoded by intron-containing genes and initiator tRNA-Met (tRNA-iMet) encoded by intronless genes was analyzed by FISH. Under normal growth conditions, the *los1∆ msn5∆* cells accumulate large amounts of mature tRNAs in the nucleus. This tRNA gradient across the NE was abolished by treating the cells with 2-deoxyglucose and NaN_3 and was re-established by removing these drugs in the presence of thiolutin, an inhibitor of all the three RNA polymerases (*Takano et al., 2005*; *Figure 2—figure supplement 1*). If Ssa2p is involved in the nuclear import of tRNAs, the *los1∆ msn5∆ ssa2∆* mutant would accumulate less mature tRNAs in the nucleus and fail to re-establish the tRNA gradient across the NE during energy recovery. As shown in *Figure 2—figure supplement 1*, neither the *ssa1∆* nor *ssa2∆* mutants exhibits altered tRNA localization in a *los1∆ msn5∆* background. Furthermore, *ssa2∆* mutant cells re-established the tRNA gradient across the NE after energy depletion like the other strains (*Figure 2—figure supplement 1*, bottom row). These results indicate that neither Ssa2p nor Ssa1p plays a major role in tRNA import under normal growth conditions.

Then, an effect of the *ssa2* mutation on tRNA import was assessed under the conditions of nutrient starvation. Wild-type, *ssa1Δ*, and *ssa2Δ* cells cultured in the YPD medium were transferred to the amino acid-starvation medium SD+Ura, Ade (SD), and the localization of tRNA-Pro$_{UGG}$ and tRNA-iMet was analyzed by FISH. While the wild-type and *ssa1Δ* cells accumulated both tRNA-Pro$_{UGG}$ and tRNA-iMet in the nucleus under these conditions within 2 hr, relatively lower amounts of tRNA-Pro$_{UGG}$ and tRNA-iMet were observed in the *ssa2Δ* nuclei (*Figure 2A*). We also examined three other tRNA species, tRNA-Lys$_{CUU}$, tRNA-Lys$_{UUU}$, and tRNA-Tyr$_{GUA}$, in FISH, and found that all the tRNAs were apparently affected by *ssa2Δ* mutation while the effects of *ssa1Δ* mutation were not obvious if any (*Figure 2—figure supplement 2*). The extent of nuclear accumulation under starvation conditions and the effect of *ssa2Δ* were variable among tRNA species (*Figure 2* and its supplement). To confirm difference of the two *ssa* mutations in tRNA accumulation upon starvation more in detail, quantitative analyses of the FISH images were carried out (*Figure 2A*, table). While the average nuclear accumulation indices (NAIs), rations of nuclear FISH signals against cytosolic signals, of tRNA-Pro$_{UGG}$ in the wild-type, *ssa1Δ*, and *ssa2Δ* cells were nearly equal when the cells were grown in YPD, the NAIs of these strains became $1.52 \pm 0.09$, $1.58 \pm 0.10$, and $1.14 \pm 0.25$, respectively, when the cells were starved, revealing an apparent difference between tRNA import in the wild-type cells and that in the *ssa2Δ* cells. We noticed that, under the starvation conditions, variation of NAIs of individual cells increased if compared with that in the rich medium (*Figure 2—figure supplement 3*) while average NAIs of biological replicates fell into a narrower range. Deletion of either of the other two *SSA* genes, *SSA3* and *SSA4*, which are only expressed under stress conditions, had no impact on tRNA accumulation under starvation conditions (*Figure 2A*, table). When the effect of simultaneous deletion of *SSA1* and *SSA2* genes was examined, the double deletion did not result in the obvious additive effect (*Figure 3*). Similar results were obtained when distribution of tRNA-iMet was quantified (data not shown).

An importin β, Mtr10p, was shown to participate in tRNA redistribution under starvation conditions (*Shaheen and Hopper, 2005*). Thus, we examined whether Ssa2p acts in the same pathway as Mtr10p, using an *ssa2Δ GAL7p-MTR10* double mutant strain. As some controversy exists regarding the involvement of Mtr10p in the tRNA import (*Shaheen and Hopper, 2005*; *Chafe et al., 2011*), the promoter shut-off strain was used to circumvent any possible adaptation effects due to the *MTR10* deletion, which causes a strong growth defect. These strains were grown in YPGal medium and then grown in YPD for 18 hr to shut-off the expression of *MTR10* before being subjected to nutrient starvation. As reported by *Shaheen and Hopper (2005)*, nuclear accumulation defects were evident in the *MTR10* shut-off strain (*Figure 2B*, *mtr10↓*). The *MTR10* shut-off had a stronger effect on tRNA distribution than the *ssa2Δ* single mutation. The double mutant showed an additive decrease in the signal intensity of nuclear tRNA-Pro$_{UGG}$ (*Figure 2B*, *ssa2Δ mtr10↓*), and signal quantification revealed that the average NAI of the starved double mutant ($0.87 \pm 0.04$) was clearly below those observed for the *mtr10* and *ssa2Δ* single mutants ($0.98 \pm 0.04$ and $1.11 \pm 0.07$, respectively), and was far below that observed for the wild type strain ($1.30 \pm 0.10$). Student's *t*-test indicated that the probability that the NAIs of the *mtr10* single and *mtr10 ssa2Δ* double mutants were the same was only 0.037 while the difference between those of the *mtr10* single and *mtr10 ssa1Δ* double mutants was not significant ($p = 0.92$). Similar quantitative results were obtained for tRNA-iMet (data not shown). These results support the idea that Ssa2p and Mtr10p act independently in parallel pathways for tRNA import under starvation conditions.

If the nuclear import of tRNAs by Ssa2p is up-regulated under starvation conditions, localization of Ssa2p may be affected by starvation. Thus, localization of FLAG-tagged Ssa proteins was monitored by immunofluorescence. While clear exclusion of Ssa1p and Ssa2p from the nucleus was observed both in rich and poor media, a slight but distinct increase in the nuclear signals of both Ssa proteins was observed when the cells were incubated in SD+Ura, Ade (*Figure 2C*). In summary, these results indicate that Ssa2p plays a pivotal role in the nuclear import of tRNAs under starvation conditions, while the other Ssa proteins, including Ssa1p, have a minor, if not at all, role in this process. Furthermore, Ssa2p may provide a novel nuclear transport pathway independent of Mtr10p.

## In vitro properties of Ssa proteins as tRNA binding proteins

To investigate biochemical characteristics of the interaction between Ssa proteins and tRNAs, we examined the ability of recombinant Ssa proteins to bind tRNAs directly in vitro. Interactions between

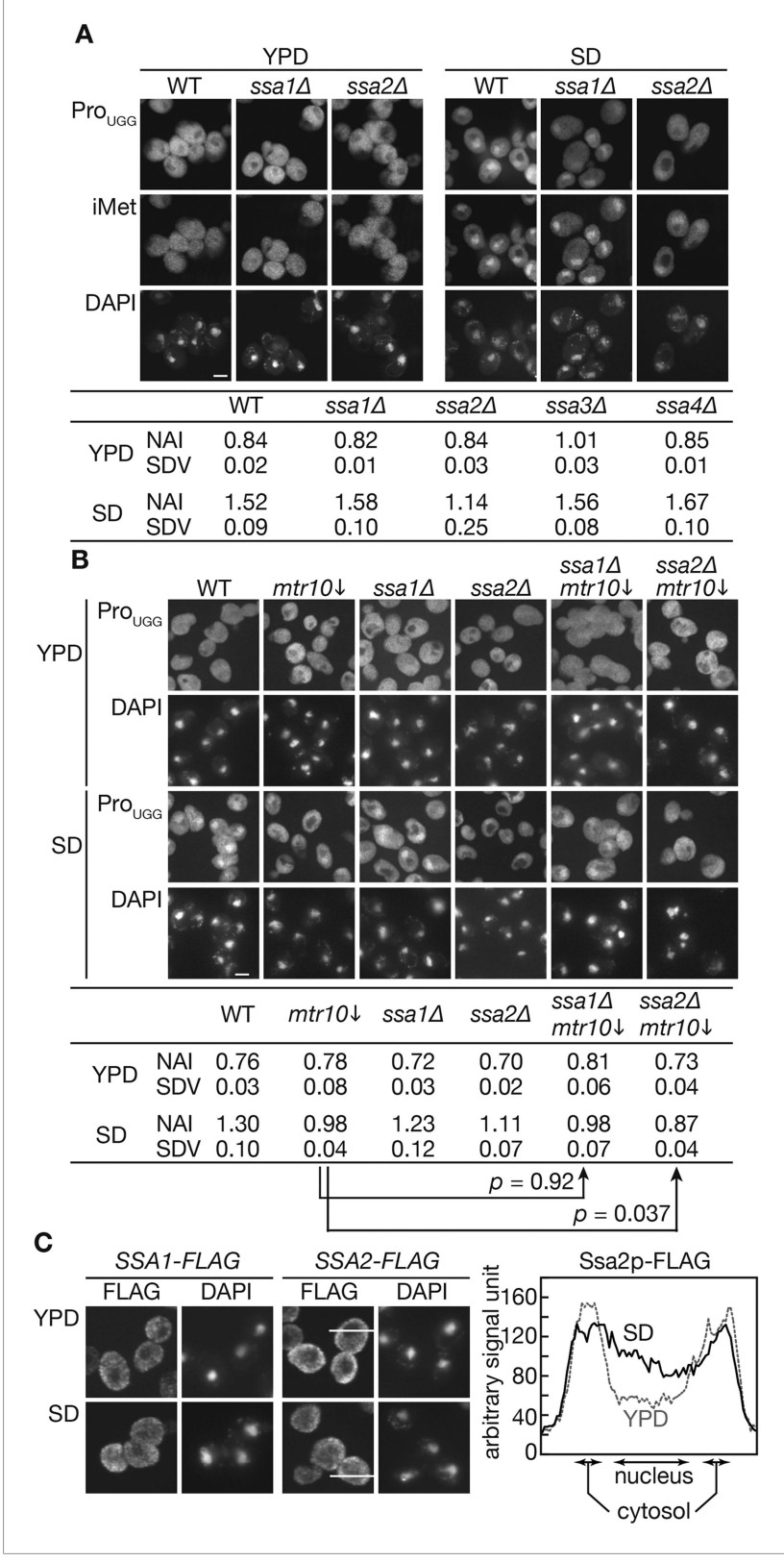

**Figure 2**. Nuclear accumulation of tRNAs under starvation conditions is affected by *SSA2* gene deletion. (**A**) Wild-type (WT: W303-1A), *ssa1Δ* (TYSC918), and *ssa2Δ* (TYSC920) cells were cultured in YPD until the log phase. The cells were then transferred to SD+Ura, Ade lacking all amino acids and cultured for additional 2 hr. The cells before (YPD)

*Figure 2. continued on next page*

*Figure 2. Continued*

and after amino acid-starvation (SD) were subjected to FISH with a rhodamine-labeled probe against mature tRNA-Pro$_{UGG}$ (Pro$_{UGG}$) and an FITC-labeled probe against tRNA-iMet (iMet). The nucleus was visualized with DAPI. Fluorescence signals of FISH images of tRNA-Pro$_{UGG}$ were quantified, and the ratio between the nuclear and cytosolic signals is expressed as the NAI (see 'Materials and methods'). Three independent samples were analyzed, and their average NAIs (NAI) with standard deviations (SDV) are shown. Bar, 5 μm. (**B**) The wild-type, GAL7p-MTR10 (mtr10↓: TYSC612), ssa1Δ (TYSC918), ssa1Δ GAL7p-MTR10 double mutant (ssa1Δ mtr10↓: ssa1Δ mtr10), ssa2Δ (TYSC920), and ssa2Δ GAL7p-MTR10 double mutant (ssa2Δ mtr10↓: ssa2Δ mtr10) strains pre-grown in YPGal were cultured in YPD for 18 hr, transferred to SD+Ura, Ade, and subjected to FISH with the anti-tRNA-Pro$_{UGG}$ probe, as described in **A**. Average NAIs with SDVs from three independent experiments are shown in the table.
(**C**) Localization of Ssa1p and Ssa2p was analyzed by immunofluorescence microscopy. Cells expressing SSA1-FLAG or SSA2-FLAG were grown in YPD (YPD) and were subsequently cultured in SD+Ura, Ade (SD) for 2 hr. The signal intensities of Ssa2p-FLAG on the lines shown in the pictures are shown in the far right graph. A solid line indicates a cell in SD, and a dashed gray line does a cell in YPD. Original microscopic images and individual data for quantitative FISH in this figure will be found in *Figure 2—source data 1*.

The following source data and figure supplements are available for figure 2:

**Source data 1**. Zip file containing source data for *Figure 2*.

**Figure supplement 1**. Nuclear import of tRNA under the los1Δ msn5Δ background was not affected by the deletion of SSA genes.

**Figure supplement 2**. Defects in nuclear accumulation of several other tRNA species were observed in ssa2Δ cells.

**Figure supplement 3**. Quantitative analyses of nuclear accumulation of tRNAs.

mammalian Hsc70 and short-lived RNAs with the AU-rich element were previously demonstrated by the label transfer assay (*Henics et al., 1999*). A similar assay was first employed to test whether yeast Ssa proteins bind the AU-rich element in vitro. Both Ssa1p and Ssa2p received radioactivity from $^{32}$P-labeled (AUUU)$_5$ RNA but not from (ACCC)$_5$ RNA (*Figure 4—figure supplement 1*), and label transfer was dependent on UV-irradiation and sensitive to ATP. Besides, no label transfer was observed when BSA was used as a control protein. These results demonstrate that the yeast Hsp70s have the ability to bind certain RNA molecules.

Next, tRNAs were used as label transfer substrates. Both Ssa1p and Ssa2p received radioactivity from $^{32}$P-labeled tRNA-Pro$_{UGG}$ in an ATP-sensitive manner (*Figure 4A,B*) with only a marginal difference between the capacity of the two Ssa proteins to recognize the tRNA in vitro. This is in marked contrast to the effects of ssa1Δ and ssa2Δ mutations on tRNA import in vivo. Because a tRNA possesses an unpaired adenosine on its 3′ terminus, this adenosine might be recognized by Ssa proteins as an analogue of ADP or ATP, which is usually bound by the nucleotide-binding cleft of Hsp70s. To exclude this possibility, label transfer was examined using CCA-less tRNA-Pro$_{UGG}$, which starts with the 5′ guanosine and ends with the 3′ cytidine. As shown in *Figure 4A*, both Ssa1p and Ssa2p received radioactivity efficiently from the CCA-less tRNA. Label transfer with an intron-containing pre-tRNA was also performed and revealed that Ssa proteins bind tRNAs irrespective of their anticodon loop structure. When the specificity of tRNA recognition by Ssa proteins was tested by competition experiments with short RNAs, chemical amounts of in vitro-transcribed tRNA-Pro$_{UGG}$ efficiently competed with the radioactive amount of tRNA-Pro$_{UGG}$ for label transfer (*Figure 4C*, lanes 'tRNA') while only limited competition was observed with single-stranded homo-oligo-ribonucleotides (A$_{30}$, U$_{30}$, and G$_{30}$) or a double-stranded homo-oligomer (A-U)$_{30}$. These results indicate that Ssa proteins specifically recognize tRNAs in an ATP-sensitive manner via a mechanism that is not mediated through the 3′ end adenosine or the anti-codon loop of tRNAs, and that these properties resemble those observed in vivo.

When different isoacceptor tRNAs were used as donors for radioactivity, the label transfer efficiencies were variable. In particular, tRNA-Pro$_{UGG}$ was a potent substrate for label transfer while tRNA-Phe$_{GAA}$ was a very poor substrate irrespective of $^{32}$P-labeled nucleotides used for in vitro transcription (*Figure 4D*). These results suggest that Ssa proteins bind only a subset of tRNAs, or that appropriate positioning of radiolabeled nucleotides on a tRNA molecule is essential for efficient label

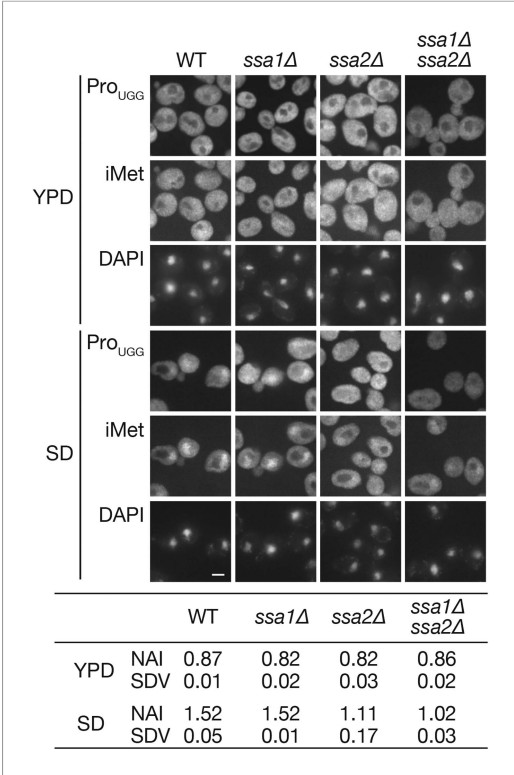

| | | WT | ssa1Δ | ssa2Δ | ssa1Δ ssa2Δ |
|---|---|---|---|---|---|
| YPD | NAI | 0.87 | 0.82 | 0.82 | 0.86 |
| | SDV | 0.01 | 0.02 | 0.03 | 0.02 |
| SD | NAI | 1.52 | 1.52 | 1.11 | 1.02 |
| | SDV | 0.05 | 0.01 | 0.17 | 0.03 |

**Figure 3**. The *ssa1Δ ssa2Δ* double deletion does not cause a synergistic effect on the nuclear accumulation of tRNAs under starvation conditions. Wild-type (W303-1A), *ssa1Δ* (TYSC918), *ssa2Δ* (TYSC920), and *ssa1Δ ssa2Δ* double mutant (TYSC1013) strains were cultured in YPD (YPD) and transferred to SD+Ade, Ura (SD) for 2 hr. The cells were subsequently subjected to FISH analysis with anti-tRNA-Pro$_{UGG}$ and tRNA-iMet probes. Bar, 5 µm. The fluorescence signals of tRNA-Pro$_{UGG}$ images of three independent experiments were quantified, and the average NAIs with SDVs were calculated. Original microscopic images and individual data for quantitative FISH in this figure will be found in *Figure 3—source data 1*.

The following source data is available for figure 3:

**Source data 1**. Zip file containing source data for *Figure 3*.

transfer. To distinguish between these two possibilities, competition experiments were performed with radiolabeled tRNA-Pro$_{UGG}$ and various non-labeled tRNAs. These experiments allowed us to examine the difference in affinity between unmodified and fully-modified tRNAs, as well. Chemical amounts of tRNAs were transcribed in vitro, and several isoacceptor tRNAs were purified from yeast by chaplet column chromatography (*Suzuki and Suzuki, 2007*). Competition experiments revealed that all tRNAs, including tRNA-Phe$_{GAA}$, competed with radiolabeled tRNA-Pro$_{UGG}$ for label transfer even though their competition efficiencies were variable (*Figure 4E*). It should be noted that unmodified tRNAs-Pro$_{UGG}$ was a more potent competitor than the fully-modified tRNA. These results indicate that various tRNAs are recognized by Ssa proteins with different affinities and that certain nucleotide positions of a tRNA molecule may offer the sites for this recognition. The above results also show that Ssa proteins recognize differences in structural characteristics between unmodified and fully-modified tRNAs.

We next examined if, in addition to the primary sequence, the higher-order structure of tRNAs is required for efficient label transfer to Ssa proteins. For this purpose, we first used hybridization of the tRNA substrate with an antisense oligonucleotide to disrupt the tRNA secondary structure. Indeed, an antisense oligonucleotide against the 5′ half of tRNA-Pro$_{UGG}$, but not an unrelated oligonucleotide, inhibited the label transfer only when hybridized with the substrate by denaturation and annealing (*Figure 5A*). Heat denaturation and quick cooling of the tRNA substrate alone did not affect label transfer efficiency, probably due to rapid refolding of the tRNA molecule. The above possibility was further tested with tRNA-Pro$_{UGG}$ mutants that exhibit altered tRNA structures (*Figure 5B*). Introduction of G18A and U54C mutations, which disrupts the interaction between the D and TΨC loops, reduced label transfer efficiency. On the other hand, destabilization of the acceptor stem by introducing G69C or G68C substitutions resulted in higher label transfer efficiency, while a C67G mutation reduced the efficiency of label transfer. Both of these positive and negative effects induced by acceptor stem mutations were minimized by compensatory mutations (*Figure 5B*, gray bars). Similar partial destabilization of the acceptor stem of other tRNA species is recognized as the degradation flag by CCA transferase to introduce an unusual CCACCA sequence instead of the normal CCA to the 3′-end of the tRNAs (*Wilusz et al., 2011*). We tested whether Ssa proteins also prefer such mutant tRNAs. Indeed, replacement of the acceptor stem of human tRNA-Leu$_{AAG}$ with acceptor-like stems from unstable tRNA-like noncoding RNAs (mouse MALAT1-associated cytoplasmic small RNA and MEM β RNA) enhanced recognition by Ssa2p (*Figure 5C*). This is also true even for single replacement of the G1-C71 Watson-Crick pair with the non-Watson-Crick pair (G1•U71) in tRNA-Arg$_{UGC}$. These results suggest that Ssa proteins recognize the overall structure of tRNAs, and prefer tRNAs with a destabilized acceptor stem but with a stable core structure.

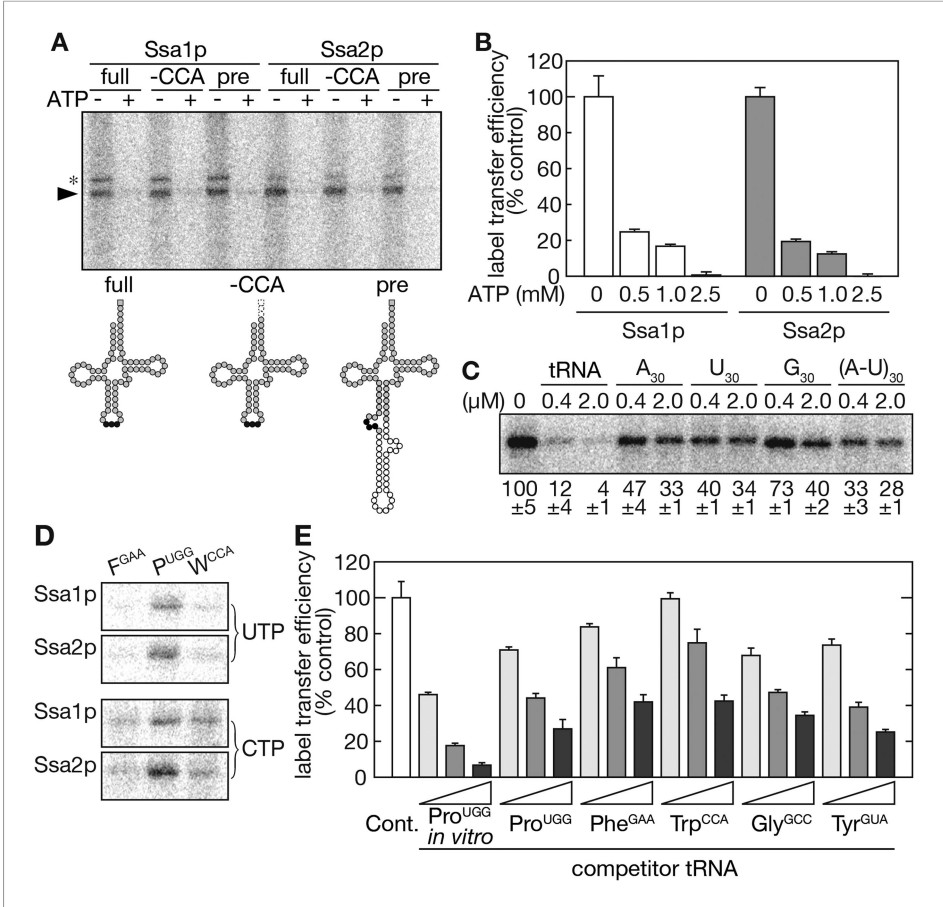

Figure 4. Ssa proteins directly and specifically interact with tRNAs in vitro. (A) In vitro-transcribed tRNA-Pro$_{UGG}$ molecules shown in the lower panel (full-length [full], CCA-less [-CCA], and precursor [pre]) were subjected to label transfer assays with recombinant Ssa1p or Ssa2p in the absence (−) or presence (+) of 2.5 mM ATP. $^{32}$P-labeled Ssa proteins are indicated by an arrowhead. Bands marked with an asterisk are Ssa proteins cross-linked with partially digested tRNA species. In the lower part, nucleotides corresponding to the anticodon, the intron and the adenosine of the CCA end are shown as black circles, white circles, and a gray square, respectively. (B) Label transfer assays with $^{32}$P-labeled mature tRNA-Pro$_{UGG}$ were performed in the presence of various concentrations of ATP. The label transfer efficiency without ATP was set to 100%. Data are presented as the averages of three independent experiments. Error bars represent SDVs. (C) The specificity of tRNA recognition by Ssa2p was examined by competition experiments with short RNAs. The indicated concentrations of in vitro-transcribed tRNA-Pro$_{UGG}$ or homo-30mers shown on the top were added to the label transfer assays. Averages of relative band intensities in triplicate experiments are shown at the bottom with SDVs. (D) Label transfer assays were performed with equal radioactive amounts of three different tRNAs (tRNA-Phe$_{GAA}$, tRNA-Pro$_{UGG}$, and tRNA-Trp$_{CCA}$) labeled with either $^{32}$P-UTP (upper) or $^{32}$P-CTP (lower). (E) Label transfer from $^{32}$P-UTP-labeled tRNA-Pro$_{UGG}$ to Ssa2p was monitored in the presence of increasing amounts of competitor tRNAs transcribed in vitro or purified from yeast. Label transfer efficiency without any competitor (white bar) was set to 100%. Original gel images and individual quantification data for the label transfer assays in this figure will be found in *Figure 4—source data 1*.

The following source data and figure supplement are available for figure 4:

**Source data 1**. Zip file containing source data for *Figure 4*.

**Figure supplement 1**. Label transfer from the AU-rich RNA to Ssa proteins.

## The NBD of Ssa proteins is responsible for tRNA recognition

Hsp70 is composed of an N-terminal nucleotide-binding domain (NBD), a substrate-binding domain (SBD), and a C-terminal variable domain (CVD) (*Figure 1—figure supplement 1*). Hsp70 binds

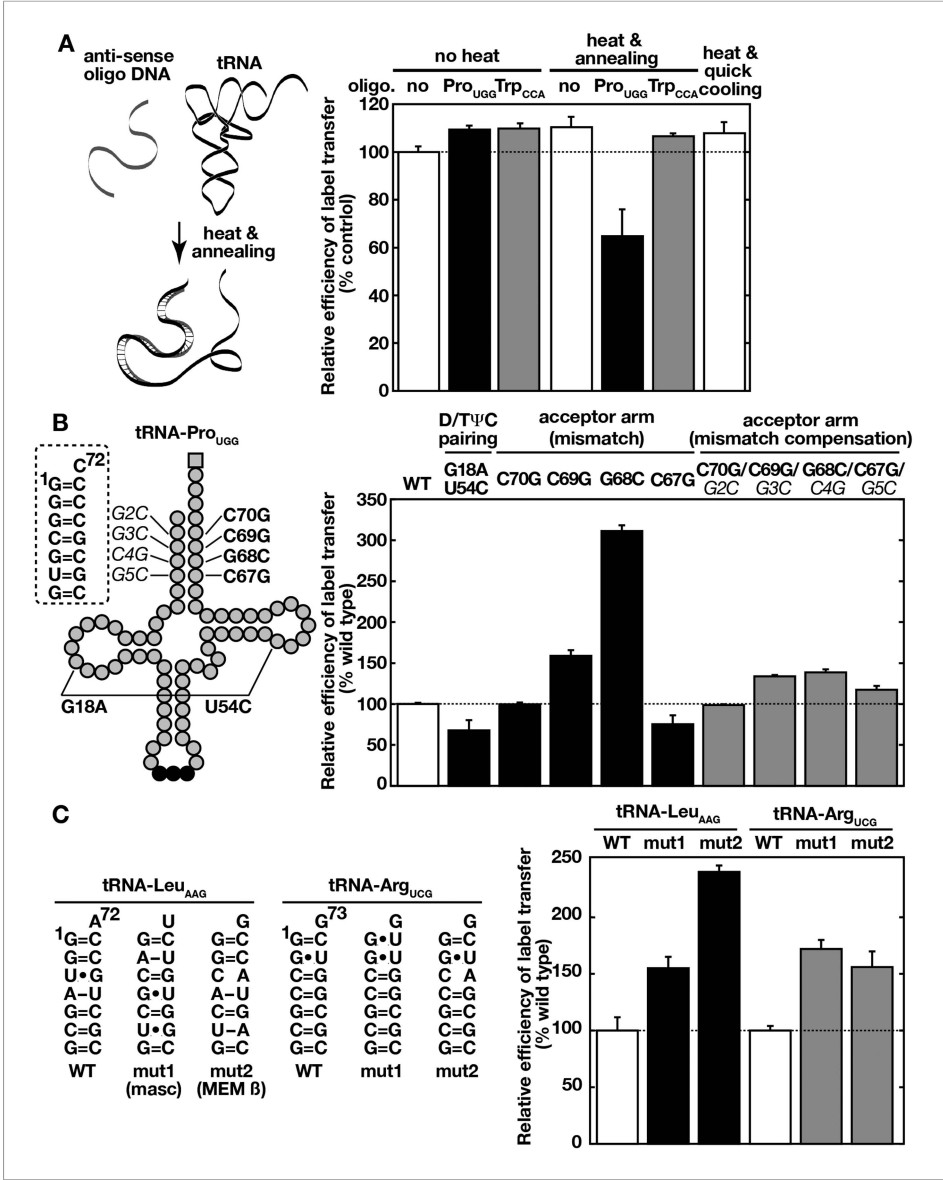

**Figure 5**. Ssa proteins recognize higher order structures of tRNAs and prefer a destabilized acceptor stem. (**A**) $^{32}$P-labeled tRNA-Pro$_{UGG}$ was mixed without (no, white bar) or, with its anti-sense (Pro$_{UGG}$, black bar) or unrelated control (anti-sense of Trp$_{CCA}$, gray bar) oligo-DNA. tRNA/oligo-DNA mixtures were heat-denatured and annealed to form DNA-RNA hybrids ('heat and annealing') as indicated on the left. A sample without oligo-DNA was also heated and quick-cooled (the far right bar). The resulting samples were then subjected to the label transfer assay with Ssa2p. (**B**) Label transfer from tRNA-Pro$_{UGG}$ mutants was assayed with Ssa2p. Mutation sites of tRNA-Pro$_{UGG}$ used in the assay are indicated schematically in the left. Boldface characters indicate mutations introduced to cause destabilization, and italicized characters do mutations that compensate for mismatches caused by the former mutations. The label transfer efficiencies of mutant tRNAs to Ssa2p were summarized in the right graph. (**C**) Wild-type (WT) and mutant forms (mut1 and mut2) of human tRNA-Leu$_{AAG}$ and tRNA-Arg$_{UCG}$ were subjected to the label transfer assay with Ssa2p. Sequences of the acceptor stems of the tRNAs are shown in the left. The tRNA-Leu$_{AAG}$ derivatives received replacements of the acceptor stem of tRNA-Leu$_{AAG}$ with those of tRNA-like ncRNAs such as MALAT1-associated small cytoplasmic RNA (mascRNA, mut1) and MEM β RNA (mut2). For panels **B** and **C**, the label transfer efficiency of the wild-type tRNA was set to 100%. Similar results were obtained with Ssa1p (not shown). Original gel images and individual quantification data for the label transfer assays in this figure will be found in *Figure 5—source data 1*.

The following source data is available for figure 5:

**Source data 1**. Zip file containing source data for *Figure 5*.

proteinaceous substrates through its SBD when its NBD binds ADP, and releases them when the NBD binds ATP (*Young et al., 2004*; *Kampinga and Craig, 2010*). Thus, we examined whether Ssa proteins utilize their SBD for recognition of tRNAs. In the label transfer assay, increasing amounts of reduced and carboxymethylated lactalbumin (RCMLA), a model unfolded protein often used in chaperone assays, were added up to 10-fold molar excess of Ssa proteins. As shown in *Figure 6A*, RCMLA addition did not inhibit, but rather moderately enhanced label transfer from tRNA-Pro$_{UGG}$. These results suggest that tRNAs are not recognized by Ssa proteins through their SBDs. To examine this notion further, purified glutathione-S transferase (GST)-fusions encompassing different Ssa domains were subjected to the label transfer assay. GST-NBD and GST-NBD-SBD fusions of Ssa1p and Ssa2p received radioactivity from tRNA-Pro$_{UGG}$ while fusions without NBD did not, indicating that the NBD plays an essential role in tRNA recognition by Ssa proteins (*Figure 6B*). It should be noted that the label transfer efficiencies of the GST-NBD and GST-NBD-SBD proteins were apparently lower than that of full-length Ssa proteins (*Figure 6B*, the left most 6 lanes). Therefore, the CVD appears to contribute to the efficient recognition of tRNAs by the NBD.

Since many mutations of Hsp70s affecting their structures and/or functions have been identified in their NBD (*McClellan and Brodsky, 2000*; *Chang et al., 2010*), we further examined whether such mutations alter the tRNA binding ability of the NBD of Ssa proteins. We introduced several mutations altering conserved amino acid residues in the nucleotide-binding cleft to GST-Ssa-NBD fusions (*Figure 6—figure supplement 1*). Mutations affecting efficiency of ATP hydrolysis and its activation, namely F66L and K69Q, enhanced tRNA binding while those compromising nucleotide binding/ exchange (G199D) reduced the binding (*Figure 6C*). Those mutations in the NBD that alter structural response to the nucleotide state (E228Q, D229N) caused virtually no changes in the tRNA binding. These results indicate that particular residues in the NBD are responsible for tRNA recognition by Ssa proteins, and that the structural features of the ATP/ADP binding state are important for the tRNA binding. Interestingly, the NBDs of Ssa1p and Ssa2p responded differently to these mutations; the Ssa1p-NBD responded more strongly to F66L and K69Q than the Ssa2p-NDB. More prominently, E173S and T201A, which affect the nucleotide binding/exchange, only caused negative effects on the Ssa2p-NBD, suggesting that these differences in in vitro-binding of tRNAs might be the reason why Ssa2p but not Ssa1p is responsible for nuclear import of tRNAs in vivo.

## Ssa proteins interact with a certain nucleoporin

If Ssa2p is directly involved in the transport of tRNAs through the NPC, Ssa2p needs to interact with nucleoporins (Nups), components of the NPC. Thus, we tested this possibility by the pull-down assay. We mixed recombinant GST, GST-Nup100(1–640)p, GST-Nup116(165–715)p or GST-Nsp1(1–601)p fusions with Ssa proteins, and recovered the GST fusions with Glutathione Sepharose beads. These portions of Nups are mostly composed of FG repeats, and are supposed to form a hydrogel phase in the NPC to allow selective transport of macromolecules (*Frey et al., 2006*; *Frey and Görlich, 2007*; *Patel et al., 2007*). In the control assays, both Ssa1p and Ssa2p were absent in the bound fraction of glutathione-beads pre-incubated with GST (*Figure 7A*). Interestingly, both Ssa proteins specifically interacted with GST-Nup116(165–715)p, but not with GST-Nup100(1–640)p or with GST-Nsp1(1–601)p. These results suggest that Ssa proteins can interact with the NPC through binding to certain Nups, such as Nup116p.

Then, we wanted to know whether Ssa proteins support NPC-interaction of tRNAs. To this end, the 3′-terminally Alexa 488-labeled tRNA-Pro$_{UGG}$ and glutathione-beads coated with GST or GST-Nup116 (165–715)p were incubated in the absence or presence of the Ssa protein, and the mixtures were subjected to fluorescence microscopy (*Patel et al., 2007*). Peripheral fluorescence on the glutathione-beads were observed only when the beads coated with GST-Nup116(165–715)p were incubated with the fluorescent tRNA in the presence of Ssa proteins (*Figure 7B*), while no such peripheral signal was seen in the absence of Ssa proteins and when GST-coated beads were used. We also performed similar experiments with Alexa 488-labled total yeast tRNAs, and obtained similar results (data not shown). These results support the idea that Ssa2p drives interaction of tRNAs to the NPC.

## Co-chaperones for Ssa2p are also involved in tRNA import

Hsp70s usually cooperate with co-chaperones to conduct their functions. Hsp40/DnaJ co-chaperones accelerate the ATPase activity of Hsp70s and pass specific substrates to corresponding Hsp70s to control their functionality in a spatiotemporal manner, while co-chaperones, GrpE, Bag-1, and Hsp110, act as nucleotide exchange factors (NEFs) (*Young et al., 2004*; *Kampinga and Craig, 2010*). To

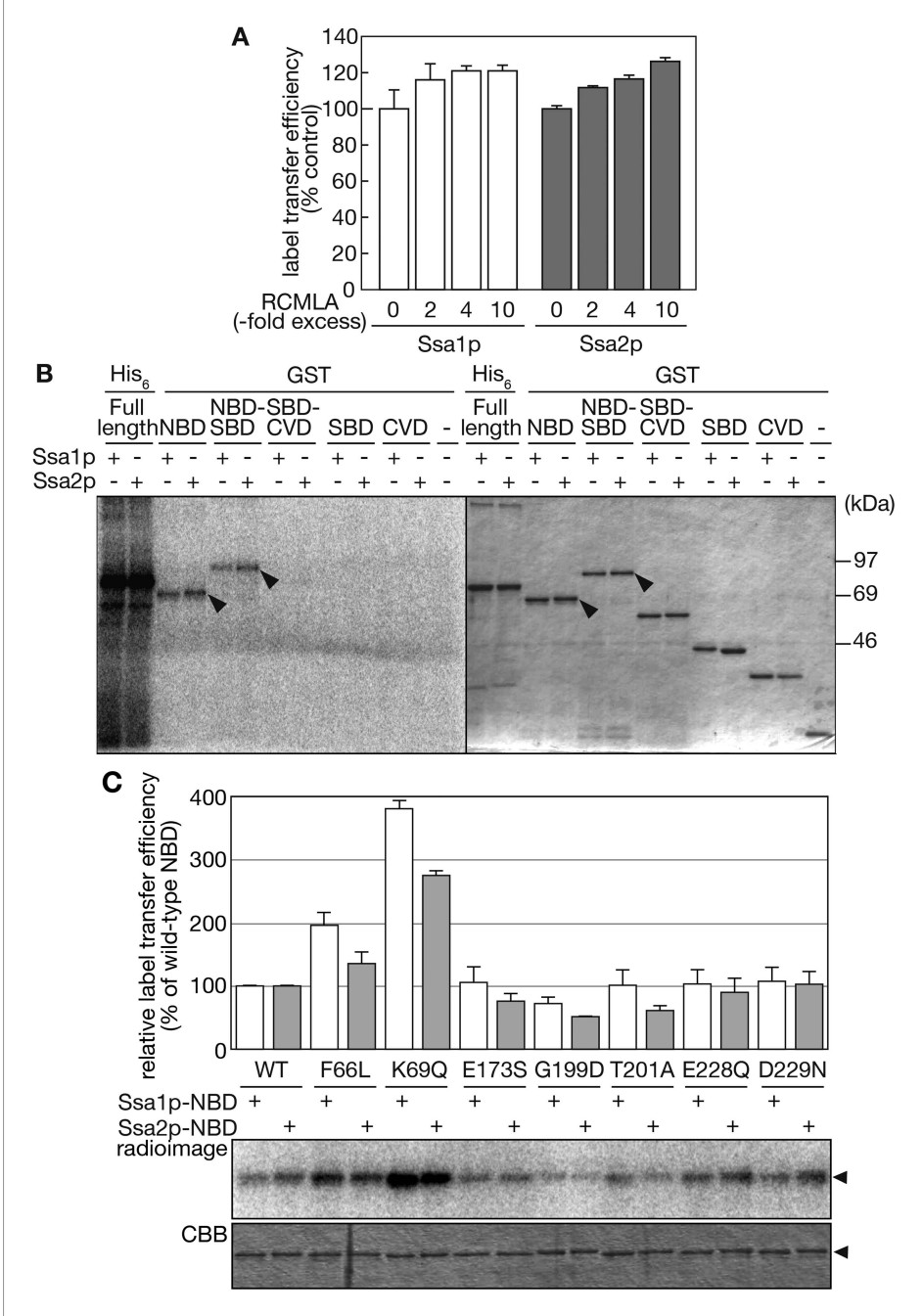

Figure 6. The NBD of Ssa proteins is essential for tRNA recognition. (A) Label transfer assays with $^{32}$P-labeled tRNA-Pro$_{UGG}$ were performed in the presence of various concentrations of RCMLA. The amount of RCMLA is shown as the molar ratio against Ssa proteins. (B) Label transfer assays with full-length Ssa-His$_6$ fusions or GST-fusions with partial Ssa proteins. Left, radioimaging; right, CBB staining. Arrowheads indicate the bands of GST-fusions that received radioactivity. (C) Label transfer assays were carried out with wild-type or mutant forms of GST-Ssa1p-NBD (Ssa1p-NBD) or GST-Ssa2p-NBD proteins (Ssa2p-NBD). Quantitated data and raw gel images of a typical experiment are shown in the upper graph and the lower panels, respectively. The relative label transfer efficiency represents a ratio of label transfer of a mutant GST-Ssa-NBD to that of the corresponding wild type. The efficiency of the wild-type protein is set to 100%. All the experiments are done in triplicates, and error bars represent SDVs. Original gel images and individual quantification data for the label transfer assays in this figure will be found in *Figure 6—source data 1*.

*Figure 6. continued on next page*

*Figure 6. Continued*
The following source data and figure supplement are available for figure 6:
**Source data 1**. Zip file containing source data for *Figure 6*.
**Figure supplement 1**. Mutations introduced into the NBD of Ssa proteins.

examine whether certain yeast homologues of DnaJ are involved in nuclear accumulation of tRNAs under starvation conditions, three major cytoplasmic DnaJ proteins, Sis1p, and Ydj1p, which are specific for Ssa proteins, and Zuo1p, which is specific for another class of cytoplasmic Hsp70s, Ssb proteins, were chosen for further analyses. As shown in *Figure 7*, both the *sis1-151* and *ydj1Δ* mutants showed an apparent defect in the nuclear accumulation of tRNAs under starvation conditions while the *zuo1Δ* mutant cells were able to accumulate tRNAs in the nucleus. This is consistent with the finding that Ssb1p and Ssb2p are excluded from the nucleus and stay in the cytoplasm while Ssa1p and Ssa2p shuttles between the two compartments (*Shulga et al., 1999*), and indicates that DnaJ proteins specific for Ssa1p and Ssa2p are required for tRNA import. If the ATPase activation of Ssa2p is required for tRNA import in the cytosol, a nuclear NEF for Ssa2p may contribute to nuclear import, as in the case of GEF for Ran GTPase (*Grünwald et al., 2011*). In yeast, such an NEF is Snl1p, a Bag-1 homologue localized on the nucleoplasmic side of the NE (*Sondermann et al., 2002*). In contrast to the case of *sis1-151* and *ydj1Δ*, *snl1Δ* did not affect nuclear accumulation of tRNAs under starvation conditions, indicating that Snl1p is not involved in this process. Nevertheless, the effects of the *sis1* and *ydj1* mutations support the assumption that the ATPase cycle of Ssa2p plays an essential role in tRNA import.

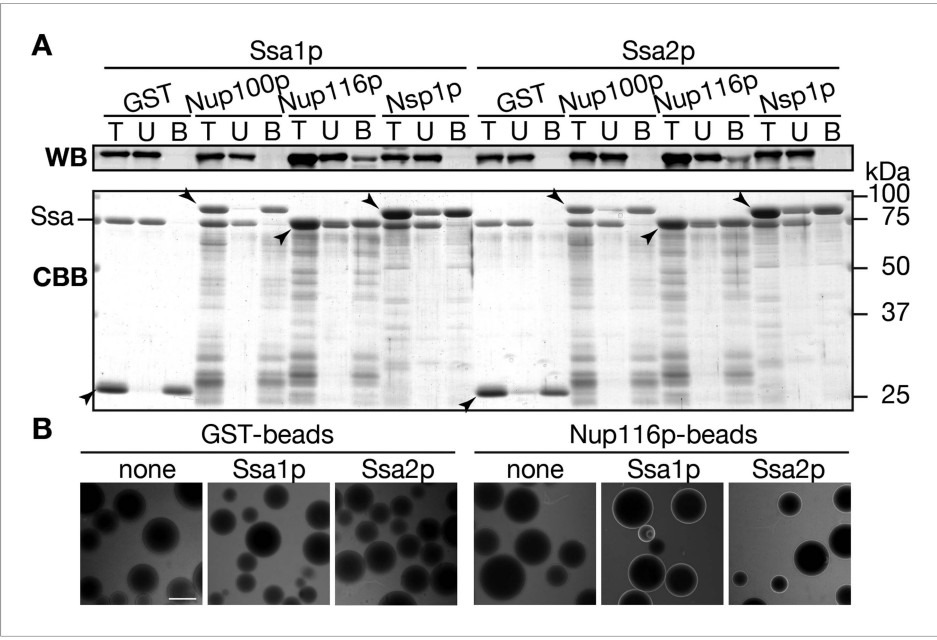

**Figure 7**. Ssa proteins interact with a certain nucleoporin. (**A**) Interaction between Ssa proteins and nucleoporins was analyzed by the pull-down assay. 200 pmol of either GST, GST-Nup100(1–640)p, GST-Nup116(165–715)p or GST-Nsp1(1–601)p were mixed with 40 pmol of either Ssa1p or Ssa2p, and were pull-downed with Glutathione Sepharose. The same portions of total (T), unbound (U), and bound (B) samples were subjected to SDS-PAGE. Ssa proteins in each fraction were detected by Western blotting with anti-Ssa protein antibodies (upper, WB), and total proteins were visualized by CBB staining (lower, CBB). Positions of GST fusions were indicated by arrowheads. (**B**) tRNA interaction with Nups were assayed by a variant of the low affinity binding assay. Alexa 488-labeled tRNA-Pro$_{UGG}$ (0.50 µg) was incubated with GST- or GST-Nup116(165–715)p-coated Glutathione Sepharose in the absence (none) or presence of Ssa1p (Ssa1p) or Ssa2p (Ssa2p). Binding of the fluorescent tRNA to the beads was monitored with a fluorescence microscope.

# Discussion

Hsp70 is a versatile Swiss Army knife for the protein world through its binding and releasing of protein substrates with destabilized structures (*Werner-Washburne et al., 1987*; *Nelson et al., 1992*; *James et al., 1997*; *Daugaard et al., 2007*; *Vos et al., 2008*). Unexpectedly, in the biochemical search for factors driving the nuclear import of tRNAs, we identified a major cytosolic Hsp70, Ssa2p, as a novel tRNA-binding protein that affects tRNA distribution under starvation conditions. Although Ssa2p has a highly homologous counterpart, Ssa1p, our findings, especially in vivo results, revealed difference between Ssa2p and Ssa1p in their involvement in nuclear import of tRNAs. And the following lines of evidence support the idea that Ssa2p has a novel function in intracellular dynamics of tRNAs as their nuclear import carrier. First, the *ssa2Δ*, but not *ssa1Δ*, mutant is defective in nuclear accumulation of the tRNAs analyzed so far under nutrient starvation (*Figure 2*). Besides, the *ssa1Δ ssa2Δ* double mutant does not show any additive defect in tRNA import (*Figure 3*). Because we have not analyzed all the isodecoder tRNAs in the yeast, there remains a possibility that nuclear import of certain tRNAs is driven by Ssa1p. Second, Ssa2p binds various tRNA species specifically and directly both in vivo and in vitro, while tRNA binding of Ssa1p was only observed in vitro (*Figures 1D, 4*). We noticed that mutations in the conserved residues in the NBD of the Ssa proteins affect the tRNA binding ability in vitro differently between Ssa1p and Ssa2p (*Figure 6C*). A possible explanation for the difference between the in vivo and in vitro results is that, in vivo, an auxiliary factor(s), such as co-chaperones regulating the ATPase cycle of Ssa2p, may confer specificity to Ssa2p as a tRNA import factor, or that the small difference in the tRNA binding of the two NBDs in vitro is enhanced to yield labor assignment between Ssa1p and Ssa2p in vivo when these multi-functional proteins are surrounded by variety of protein and RNA substrates. Third, Ssa proteins are known to shuttle between the cytoplasm and the nucleus, and their nuclear-cytoplasmic distribution is regulated according to environmental conditions (*Shulga et al., 1999*). Indeed, we observed a marginal but reproducible increase in the nuclear pool of Ssa2p-FLAG upon amino acid starvation (*Figure 2C*). Such alteration of transport carrier distribution was reported recently for Los1p, Msn5p, so on upon glucose starvation (*Huang and Hopper, 2014*). Fourth, Ssa proteins interact with the FG-repeat region of Nup116p, and support association of tRNAs to Nup116p, suggesting that Ssa2p can enter the NPC with tRNAs under certain conditions (*Figure 7*). Fifth, we demonstrated the involvement of Sis1p and Ydj1p, major cytoplasmic DnaJs for Ssa proteins, in tRNA import in vivo, while such effect is not observed in the *zuo1Δ* mutant, which is defective in Ssb-protein specific DnaJ (*Figure 8*). These results collectively suggest that tRNA loading onto Ssa2p may be assisted by these cytosolic co-chaperones specific for the Ssa proteins, and ATP hydrolysis by Ssa2p may drive upward transport of tRNAs across the NPC. Therefore, Ssa2p is the first factor that assists tRNA import by direct binding to tRNAs, shuttles between the cytosol and the nucleus, interacts with the NPC component, and couples the tRNA transport with energy release.

There might be a possibility that Ssa2p acts as an import adaptor for tRNAs, like importin-α for proteins targeted to the nucleus, and Mtr10p, an importin β, functions as an import carrier of the tRNA-Ssa2p complex. However, the *ssa2Δ* and *mtr10*-shut off double mutant exhibited an additive defect in tRNA accumulation as compared with single mutants (*Figure 2B*). While this additive effect was not so significant, the difference between the double mutant and the single mutants was statistically meaningful. Although we could not completely negate the above possibility so far, these data rather support the idea that Ssa2p and Mtr10p form parallel and independent pathways, as the case of Los1p and Msn5p in tRNA export. Another potential non-importin candidate for an Ssa2p import carrier is Opi10p, a yeast homologue of Hikeshi, which functions as a nuclear import carrier of mammalian Hsp70 under heat stress conditions (*Kose et al., 2012*). However, our preliminary experiments suggest that the *opi10Δ* mutant does not affect localization of Ssa1p or Ssa2p under normal or starvation conditions (data not shown). Although there are several other possibilities for involvement of the Ssa2p chaperone system in tRNA import, from the collective pieces of experimental results, we primarily propose that Ssa2p acts as an import carrier of tRNAs across the NPC. Since the *ssa2Δ* phenotype is only obvious under starvation conditions, Ssa2p may provide a secondary pathway to accommodate high demand for the tRNA import under these conditions. In other words, the Ssa2p pathway may confer a regulatory capability to the tRNA nuclear import, responding to growth conditions like the case of tRNA export (*Murthi et al, 2010*; *Huang and Hopper, 2014*; *Pierce et al., 2014*). We speculate that Ssa2p also has some contribution to basal

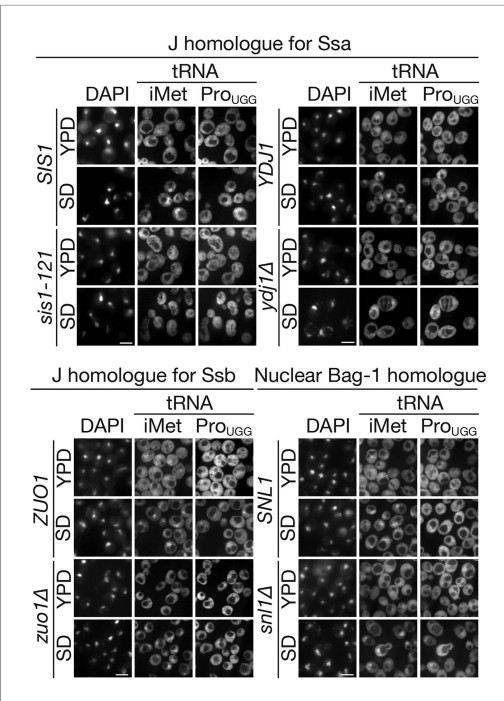

**Figure 8**. Major cytosolic DnaJ homologues of Ssa proteins are involved in tRNA import under starvation conditions. tRNA localization under starvation conditions in mutants of major cytoplasmic DnaJ homologues (*SIS1*, *YDJ1* and *ZUO1*) or a nuclear Bag-1 homologue (*SNL1*) was analyzed by FISH. Pairs of *YDJ1* (PJ31-3A) and *ydj1Δ* (JJ160) strains, *SIS1* (TYSC950) and *sis1-121* (TYSC951) strains, *ZUO1* (BY4741) and *zuo1Δ* (5937) strains, and *SNL1* (W303-1B) and *snl1Δ* (SWY1353) strains were treated as described in **Figure 2** except that all the strains but the *SNL1* and *snl1Δ* strains were cultured at 23°C instead of 30°C. In each set of panels, upper two rows are the parental wild type and the lower two rows are the mutant. Bar, 5 μm.

nuclear import of tRNAs under normal conditions by co-operating with the Mtr10p pathway, and that quantitative regulation of the Ssa2p pathway, but not complete on-and-off, enables enhancement of nuclear import under starvation conditions.

As mentioned above, we observed a direct interaction between Ssa proteins and tRNAs, which is governed by a mode different from that for recognition of protein substrates by Hsp70. tRNA binding is saturable and is competed specifically by other tRNA molecules but not by a protein substrate (**Figures 4, 6A**). We also found that tRNA binding takes place through the NBD, but not through the SBD, of Ssa proteins (**Figure 6B**). On the other hand, Ssa proteins bind tRNAs that lack an adenosine at their 3' terminus, so that a tRNA molecule is not just a mimic of monomeric adenine nucleotides (**Figure 4A**). Examination of the crystal structure of bovine Hsc70 (**Jiang et al., 2005**) indicates that the ATP-binding cleft is too narrow to accommodate the acceptor stem of the tRNA molecule while the several mutations introduced into this region affected tRNA binding of Ssa proteins, especially Ssa2p (**Figure 6C**), suggesting that the ATP/ADP-binding state of Ssa2p alters an unidentified tRNA-binding interface on the NBD.

While disruption of the overall tRNA structure decreases the tRNA affinity for Ssa proteins, some mutations on the acceptor stem enhance Ssa protein recognition (**Figure 5**). In addition, Ssa proteins prefer unmodified tRNAs to fully-modified tRNAs for their binding (**Figure 4E**). Therefore, Ssa proteins appear to recognize some characteristics of the three-dimensional structure of tRNA molecules, though preferably loosely-folded structures of tRNAs. Although the tRNA import system(s) seems to accept a variety of tRNA species, including matured and stably folded tRNAs (**Takano et al., 2005**), such preference for unmodified or loosely-folded tRNAs may well contribute to the quality control of cytosolic tRNAs. Recent studies of rapid tRNA decay (RTD), which degrades hypomodified aberrant tRNAs in the cytosol, revealed that the RTD system also recognizes tRNAs with some defects in their acceptor stem, and a part of such aberrant tRNAs are degraded by the nuclear exonuclease Rat1p (**Alexandrov et al., 2006**; **Whipple et al., 2011**; **Wilusz et al., 2011**). Because the above characteristics of the NBD of Ssa proteins are similar to those of the SBD with respect to protein binding (**Deshaies et al., 1988**; **Tanaka et al., 2002**), we postulate that the NBD possesses a chaperone-like activity for tRNAs in recognizing their structural features, and may contribute to substrate selection for the nuclear RTD through tRNA import, if such tRNAs escape from the cytoplasmic RTD.

Although there may be other potential roles of Ssa2p in nuclear import of tRNAs, binding and releasing of tRNAs by Ssa2p, which is likely coupled with transport across the NPC, could constitute an intrinsic step for the tRNA import. An attractive idea is that tRNA import is driven by the ATPase cycle of the Hsp70 in a similar manner to the nuclear import driven by the Ran GTPase cycle. The Ssa2p-mediated pathway may function as a regulatory pathway for tRNA import to adapt nutrient stress by using the heat-shock protein in a novel mode of action. Future studies should reveal the

detailed molecular mechanism, by which Hsp70 contributes to tRNA import into the nucleus and its regulation upon nutrient stress.

# Materials and methods

## Strains and plasmids

Yeast genetic techniques are essentially described in *Guthrie and Fink (1991)*, and other molecular biological techniques are in *Sambrook and Russell (2001)*. *S. cerevisiae* strains used in this study are summarized in *Supplementary file 2*.

## Preparation materials used in the experiments

### Immobilized tRNA resin

Immobilization of tRNAs on an agarose matrix was performed according to the method published by *Hermanson et al. (1992)*. In brief, 4.0 mg yeast tRNAs (Roche Diagnostics, Mannheim, Germany) were activated by treatment with 4.0 mg $NaIO_4$ at 4°C for 1 hr and then ethanol-precipitated to remove unreacted $NaIO_4$. Activated tRNAs were subsequently coupled to approximately 1 ml of a hydrazide matrix, Affi-Gel Hz (Bio-Rad Laboratories, Hercules, CA), in 0.10 M Na-acetate, pH 5.0 at 4°C for 4 hr. After extensive reciprocal washes with 0.10 M Na-acetate, pH 5.0, and 2.0 M NaCl, the resin was equilibrated with Buffer A (50 mM Tris–HCl, pH 7.4, 350 mM NaCl, 5 mM $MgCl_2$). Approximately, 1 mg/ml tRNAs were immobilized on the tRNA resin.

### Reduced and carboxymethylated lactalbumin

RCMLA was prepared essentially as described in *Ferber and Ciechanover (1986)*. Briefly, α-lactalbumin (Sigma–Aldrich, St. Louis, MO) was denatured in 0.40 M Tris–HCl, pH 8.6, 5 mM EDTA, 6.0 M guanidine-HCl, and 50 mM DTT under $N_2$ gas at 37°C for 1 hr. After addition of Na-iodoacetate to a final concentration of 100 mM, the denatured protein was incubated for 1 hr. The resulting sample was dialyzed against 10 mM K-Pi, pH 7.5, and 150 mM KCl. Carboxymethylation and denaturation were confirmed by SDS-PAGE and circular dichroism spectrometry, respectively.

### Radiolabeled RNAs

tRNA-Pro$_{UGG}$, its derivatives, a (AUUU)$_5$-containing RNA and a (ACCC)$_5$-containing RNA were cloned in pUC119 by PCR amplification of the DNA fragments or by annealing of single-stranded (ss) oligonucleotides encoding the RNAs with the T7 promoter (see *Supplementary file 1*). Template plasmids for the tRNAs and those for the other RNAs were linearized with *Bsm*AI and *Eco*RI, respectively. For preparation of DNA templates of human tRNA-Leu$_{AAG}$ and tRNA-Arg$_{UCG}$ with the T7 promoter on the 5′ side of the tRNA sequences, and their derivatives, whose sequences are described in *Wilusz et al. (2011)*, 60 nt-long partial sense and anti-sense oligonucleotides with an overlap of 18 nt are annealed, and the single-stranded regions were filled by Ex Taq (Takara Bio, Otsu, Japan). Linear DNAs were subjected to in vitro transcription with MAXIscript (Ambion, Austin, TX) in the presence of α-$^{32}$P-UTP or α-$^{32}$P-CTP (PerkinElmer, Waltham, MA). After purification of radiolabeled RNAs, the quality and radioactivity of the RNAs were analyzed by urea-PAGE and by liquid scintillation counting, respectively.

### Chemical amounts of competitor RNAs

(A)$_{30}$, (U)$_{30}$, and (G)$_{30}$ oligo-ribonucleotides were obtained from Sigma Genosys (St. Louis, MO), and an (A-U)$_{30}$ double-stranded (ds) oligo-ribonucleotide was prepared by heat-denaturing and annealing of corresponding ss oligo-ribonucleotides. In vitro transcribed tRNA-Pro$_{UGG}$ was prepared by MEGAshortscript (Ambion) using *Bsm*AI-linearized pTYE326 as a template. After DNase treatment, phenol-chloroform extraction and ethanol precipitation, residual monomeric nucleotides in the tRNA samples were further removed using NucAway spin columns (Ambion). Naturally occurring yeast tRNAs (tRNA-Pro$_{UGG}$, tRNA-Phe$_{GAA}$, tRNA-Trp$_{CCA}$, tRNA-Gly$_{GCC}$, and tRNA-Tyr$_{GUA}$) were purified from a commercial yeast tRNA mixture (Roche Diagnostics) by chaplet column chromatography (*Suzuki and Suzuki, 2007*). In brief, a biotinylated oligonucleotide against a specific isoacceptor tRNA (approximately 30 nmol) was captured by a HiTrap streptavidin HP column (1 ml volume; GE Healthcare, Sunnyvale, CA), and the column was extensively washed with Binding Buffer (30 mM HEPES-KOH, pH 7.5, 15 mM EDTA, 1.2 M NaCl). The columns for tRNA-Pro$_{UGG}$, tRNA-Phe$_{GAA}$, tRNA-Trp$_{CCA}$, tRNA-Gly$_{GCC}$, and tRNA-Tyr$_{GUA}$ were joined in series, and

100 mg of the yeast tRNA mixture was applied to the column set by the recycling flow at 0.5 ml/min at 65°C for 2 hr. The columns were disconnected, and each column was washed extensively with Binding Buffer. After washing, the bound tRNA in each column was eluted with a total of 3.0 ml of TE (10 mM Tris–HCl, pH 7.5 and 1 mM EDTA) at 65°C. After ethanol-precipitation, the tRNAs were finally dissolved in 250 μl of TE.

*Alexa 488-labeled tRNA-Pro$_{UGG}$* To prepare fluorescence-labeled tRNA-Pro$_{UGG}$, the 3′-terminus of tRNA-Pro$_{UGG}$ transcribed in vitro as above (50 μg) was activated with 50 μg of NaIO$_4$ in a 25 μl reaction at 23°C for 1 hr. After passing through a PD SpinTrap G-25 desalting column (GE Healthcare) equilibrated with 0.10 M NaOAc, pH 5.2, the resulting tRNA was coupled with 17.5 nmol of Alexa Flour 488 Hydrazide at 4°C for 4 hr (Life Technology, Carlsbad, CA). After ethanol precipitation, the labeled tRNA was further purified by urea-PAGE, and RNA concentration was measured by Qubit RNA BR Assay kit (Life Technology).

## Purification of tRNA-binding proteins

A yeast cytosolic fraction was prepared from logarithmically growing the *S. cerevisiae* strain W303-1A. First, the yeast cells were converted into spheroplasts and disrupted in 50 mM Tris–HCl, pH 7.4, 350 mM NaCl, 5 mM MgCl$_2$ supplemented with 0.5 mM PMSF, a 1/1000 volume of PIC (Roche Diagnostics), and 1.0 mM β-mercaptoethanol by vigorous agitation with glass beads. The lysate was mixed with final 0.5% wt/vol of Triton X-100, and was centrifuged at 100,000×*g* for 30 min. The recovered supernatant was passed through Q-Sepharose Fast Flow (GE Healthcare) to remove endogenous tRNAs. The flow-through fraction was applied to the tRNA-resin prepared as described previously in the presence or absence of 3 mM Mg-ATP, and bound proteins were eluted with 50 mM Tris–HCl, pH 7.4, 1.5 M NaCl, 5 mM MgCl$_2$ after extensive washing. ATP-dependent or ATP-sensitive tRNA-binding proteins were identified by peptide mass fingerprinting with a Voyager DE MALDI/TOF mass spectrometer (Applied Biosystems, Foster City, CA).

## Purification of recombinant proteins

Full-length Ssa1p and Ssa2p recombinant proteins were expressed by a pCold2 vector (Takara Bio) with a C-terminal His$_6$ tag in the *Escherichia coli* strain BL21(DE3) co-expressing trigger factor overnight at 15°C. The recombinant proteins were purified with Ni-NTA agarose (QIAGEN, Hilden, Germany), and dialyzed against 20 mM Tris–HCl, pH 7.4, 10 mM NaCl. For preparing GST fusions with partial Ssa proteins, PCR-amplified gene fragments encoding either Ssa1p-NBD (1–380), Ssa1p-NBD-SBD (1–550), Ssa1p-SBD (381–550), Ssa1p-SBD-CVD (381–642), Ssa1p-CVD (551–642), Ssa2p-NBD (1–380), Ssa2p-NBD-SBD (1–550), Ssa2p-SBD (381–550), Ssa2p-SBD-CVD (381–639) or Ssa2p-CVD (551–639) were cloned into pGEX-4T-2 (GE Healthcare). Mutant forms of GST-Ssa-NBD genes were constructed by oligonucleotide-directed mutagenesis with overlap extension. Each GST fusion protein was expressed in BL21(DE3) cells and purified with Glutathione Sepharose (GE Healthcare). The eluates were dialyzed against 20 mM Tris–HCl, pH 7.4, 10 mM KCl, 2.0 mM DTT.

To prepare GST-nucleoporin fusion proteins, DNA fragments encoding the 1–640 residues of Nup100p [Nup100(1–640)p], the 165–715 residues of Nup116p [Nup116(165–715)p], or the 1–601 residues of Nsp1p [Nsp1(1–601)p] were amplified by PCR and inserted into pGEX-4T-2. These fusion proteins in addition to GST were expressed in DH5α, and purified with Glutathione Sepharose. The resulting proteins were passed through NAP-10 desalting columns equilibrated with Buffer 88 (20 mM HEPES-KOH, pH 6.8, 2 mM Mg(OAc)$_2$, 150 mM KOAc [*Baker et al., 1988*]) supplemented with 0.10% wt/vol Tween-20.

## RNA analysis

RNAs were analyzed on 7.0 M urea/10% wt/vol polyacrylamide gels in the TBE buffer, and stained with Gel Red (Biotium, Hayward, CA). The RNAs in the gels were then transferred to charged nylon membranes (Hybond N$^+$, GE Healthcare), and specific RNAs were hybridized with an appropriate digoxigenin-labeled oligonucleotide probe produced with DIG Oligonucleotide Tailing Kit, Second Generation (Roche Diagnostics). The signal was developed with ECF (GE Healthcare) and read with Storm 860 Image Analyzer (GE Healthcare).

## FISH analysis

Yeast cells were cultured in YPD until log phase and, if required, incubated in SD supplemented with only uracil and adenine (SD+Ura, Ade) for 2 hr. The cells were pre-fixed for 15 min with formaldehyde solution, fixed with a paraformaldehyde solution for 1 hr, and then subjected to FISH sample preparation with FITC or rhodamine-labeled oligonucleotide probes as described before (*Yoshihisa et al., 2003*). In the case of DnaJ homologue mutants, some modifications were necessary to improve FISH images. Spheroplast formation was done in 0.90 M sorbitol-containing buffer with 18 µg/ml of Zymolyase 100T instead of standard 36 µg/ml Zymolyase. The resulting spheroplasts immobilized on poly-Lys-coated multiwell-slides were permeabilized with 0.10% wt/vol Triton-X100, and were directly subjected to hybridization. Fluorescence images were recorded by a confocal system CSU-10 (Yokogawa, Tokyo, Japan) with a cooled CCD camera CoolSNAP HQ2 (Photometrics, Tucson, AZ) mounted on a BX-60 fluorescence microscope (Olympus, Tokyo, Japan). The images were analyzed by Metamorph software (Molecular Devices, Sunnyvale, CA). To calculate an NAI of a yeast cell, the average signal intensities of nuclear and cytosolic regions were measured in a cell, and a nuclear/cytosolic signal ratio was calculated. The NAI of each experiment is the average of individual NAIs measured in 30 or above cells (see *Figure 2—figure supplement 3*). The average NAI and its standard deviation (SDV) are calculated from NAIs obtained from three biological replicates of the same experimental conditions.

## Label transfer assay

The label transfer assay was performed essentially as described by *Henics et al. (1999)*. A $^{32}$P-labeled RNA transcribed with $^{32}$P-UTP or $^{32}$P-CTP was incubated with an appropriate protein at 30°C for 10 min in 12 mM HEPES-KOH, pH 7.9, 15 mM KCl, 10% vol/vol glycerol, 0.20 mM DTT, 0.25 units/µl RNasin, and then UV was irradiated at 90 mJ/cm$^2$. After treatment with a 1/10 volume of RNase Cocktail (Ambion), samples were subjected to SDS-PAGE and radioimaging with Imaging Plate (Fujifilm, Tokyo, Japan) and STORM 860 Image Analyzer (GE Healthcare).

## Binding assays

For a typical pull-down assay, 200 pmol of either GST, GST-Nup100(1–640)p, GST-Nup116(165–715)p or GST-Nsp1(1–601)p and 40 pmol of either Ssa1p or Ssa2p were incubated with Glutathione Sepharose in Buffer 88 with 0.10% wt/vol Tween-20 at 4°C for 2 hr. After washing the beads with the same buffer 3-times, bound proteins were eluted with the same buffer with 10 mM reduced glutathione.

The glutathione beads binding assay coupled with microscopic observation is a variant of the low affinity binding assay developed by *Patel et al. (2007)*. The Alexa 488-labled-tRNA-Pro$_{UGG}$ was mixed with Glutathione Sepharose beads that adsorbed GST or GST-Nup116(165–715)p in advance in the buffer containing 10 mg/ml BSA and 0.50 % wt/vol 1,6-hexanediol. If indicated, final 5.0 µM Ssa1p or Ssa2p was added. The resulting mixture was observed under the fluorescence microscope.

## Acknowledgements

We thank the present and previous members of our laboratory for help and suggestions, especially Dr Shuh-ichi Nishikawa for his helpful discussions on chaperones, and Ms Mine Takezawa and Kayoko Terao for their technical assistance. We also thank Drs SR Wente, EA Craig, and J Brodsky for yeast strains.

## Additional information

### Funding

| Funder | Grant reference | Author |
|---|---|---|
| Ministry of Education, Culture, Sports, Science, and Technology (MEXT) | Grant-in-Aid for Scientific Research on Innovative Areas (RNA regulation) 23112708 | Tohru Yoshihisa |
| Japan Science and Technology Agency | Precursory Research for Embryonic Science and Technology (PRESTO) | Tohru Yoshihisa |

| Funder | Grant reference | Author |
|---|---|---|
| Japan Society for the Promotion of Science (JSPS) | Grant-in-Aid for Scientific Research (C) 22570139, for Challenging Exploratory Research 26650009 | Tohru Yoshihisa |

The funders had no role in study design, data collection and interpretation, or the decision to submit the work for publication.

### Author contributions

AT, TK, Acquisition of data, Analysis and interpretation of data; MM, Acquisition of data; TE, Analysis and interpretation of data, Drafting or revising the article; TY, Conception and design, Acquisition of data, Analysis and interpretation of data, Drafting or revising the article

## Additional files

### Supplementary files

• Supplementary file 1. Plasmids used for in vitro transcription in this study.

• Supplementary file 2. Yeast strains used in this study.

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
