## [Decision Letter]

Thank you for sending your work entitled “Cytosolic Hsp70 and co-chaperones constitute a novel system for tRNA import into the nucleus” for consideration at *eLife*. Your article has been favorably evaluated by James Manley (Senior editor), Karsten Weis (Reviewing editor), and two reviewers.

The Reviewing editor and the reviewers discussed their comments before we reached this decision, and the Reviewing editor has assembled the following comments to help you prepare a revised submission.

In yeast and vertebrate cells, tRNAs in the cytoplasm traffic to the nucleus and, if nutrient status permits, the imported tRNAs again reenter the cytoplasm via RNA re-export. Although there is significant information regarding tRNA initial and re-export mechanics, the gene products involved in tRNA nuclear import remain largely unknown. In this manuscript, the authors identify the cytosolic Hsp70s, Ssa1 and Ssa2 as tRNA binding proteins and demonstrate that the function of Ssa2 is required for tRNA import upon nutrient starvation. The reviewers agreed that the biochemistry in this manuscript is strong and that it contains potentially important finding that should be published. However, there was also a consensus that revisions would be necessary prior to publication, although the extent of the revisions was debated. A key discussion point amongst the three reviewers was whether there is enough evidence to show that Ssa2 functions directly in tRNA import and whether the authors can rule out more indirect roles of Ssa2.

Major points:

1) Ssa2 as a direct tRNA importer. At this point the authors cannot entirely rule out a more indirect role of Ssa2 in tRNA import. This is a major conclusion of the paper and thus this conclusion should be strengthened. This could be done by experiments testing either whether Ssa2 shuttles between the cytoplasm and the nucleus, whether Ssa2 enters the nucleus in the absence of Ran, whether a Ssa2/tRNA complex binds FG nucleoporins or even whether Ssa2 alone or in complex with tRNA translocates to the nucleus in in vitro transport assays (using mammalian cells). Any one of these experiments (or alternatives) that would provide evidence that Ssa2 is a direct transport mediator would significantly strengthen the case and the paper.

2) tRNA binding. Unfortunately, tRNA in vitro and in vivo binding do not entirely match because both Ssa1 and Ssa2 bind tRNAs in vitro whereas as only Ssa2 does so in vivo. Whereas it is quite possible that factors are missing in vitro that restrict Ssa1 from binding tRNA in vivo, the authors should be cautious. First, the authors show that the label transfer assay allows both Ssa1 and Ssa2 to bind non-tRNA substrates (Figure 4—figure supplement 1). Second, only a single tRNA is reported for the in vivo pull-down studies and only 2 tRNAs are monitored in vivo. Regarding tRNA FISH, Figure 2 (SD *ssa1Δ*) shows an affect on tRNAPro nuclear accumulation whereas accumulation is *ssa1Δ* independent in the statistics, Figure 2 and Figure 3; is there a problem with Figure 2?

3) Do Ssa2 and Mtr10 independently affect tRNA subcellular dynamics? The authors conclude that Mtr10 and Ssa2 act via separate tRNA nuclear pathways because *ssa2Δ mtr10Δ* cells are more defective in tRNA nuclear import than either mutant alone. However, this effect is subtle (Figure 3) and not entirely convincing. Perhaps the argument could be bolstered if Figure 2—figure supplement 1 were of higher quality as an interpretation of the data is that Ssa2 is important for import upon aa starvation, in contrast to Mtr10 that is thought to affect tRNA nuclear import constitutively. In the absence of additional direct documentation for nutrient-dependent tRNA nuclear import, the authors should be cautious not to conclude that Ssa2 functions in tRNA nuclear import upon aa starvation because Ssa2 could be involved in constitutive import and tRNA nuclear accumulation upon nutrient deprivation could be due to decreased re-export to the cytoplasm.

4) Role of co-chaperones? Figure 7 is not of the same high quality as the FISH studies in the other figures. It is not possible to conclude from the images that the results are different for *sis1-121Δ* and *ydj1Δ* than *zuo1Δ*.

---

## [Author Response]

*In yeast and vertebrate cells, tRNAs in the cytoplasm traffic to the nucleus and, if nutrient status permits, the imported tRNAs again reenter the cytoplasm via RNA re-export. Although there is significant information regarding tRNA initial and re-export mechanics, the gene products involved in tRNA nuclear import remain largely unknown. In this manuscript, the authors identify the cytosolic Hsp70s, Ssa1 and Ssa2 as tRNA binding proteins and demonstrate that the function of Ssa2 is required for tRNA import upon nutrient starvation. The reviewers agreed that the biochemistry in this manuscript is strong and that it contains potentially important finding that should be published. However, there was also a consensus that revisions would be necessary prior to publication, although the extent of the revisions was debated. A key discussion point amongst the three reviewers was whether there is enough evidence to show that Ssa2 functions directly in tRNA import and whether the authors can rule out more indirect roles of Ssa2*.

*Major points*:

*1) Ssa2 as a direct tRNA importer. At this point the authors cannot entirely rule out a more indirect role of Ssa2 in tRNA import. This is a major conclusion of the paper and thus this conclusion should be strengthened. This could be done by experiments testing either whether Ssa2 shuttles between the cytoplasm and the nucleus, whether Ssa2 enters the nucleus in the absence of Ran, whether a Ssa2/tRNA complex binds FG nucleoporins or even whether Ssa2 alone or in complex with tRNA translocates to the nucleus in in vitro transport assays (using mammalian cells). Any one of these experiments (or alternatives) that would provide evidence that Ssa2 is a direct transport mediator would significantly strengthen the case and the paper*.

Thank you for your valuable comments. We also noticed that there are several possibilities of indirect involvement of Ssa2p in nuclear import of tRNAs, such as a substrate selector or mediator for import carrier loading, etc*.*, but did not mention these in the original version in detail to clarify our main interpretation. As suggested by the reviewers, we tested whether Ssa proteins can interact with nucleoporins, and allow tRNAs to interact with nucleoporins by solution binding assays. As shown in the new Figure 7 and corresponding description in Results “Ssa proteins interact with a certain nucleoporin”, we could demonstrate that Ssa proteins indeed can interact with the FG-repeat region of Nup116p by the conventional pull-down assay with GST fusions while no positive data were obtained with Nup100p and Nsp1p in addition to the control, GST alone. We could not confirm Ssa2-dependent interaction between Nup116p and tRNAs by the pull-down assay, probably from rather high off-rate of tRNAs from Ssa proteins. However, a variant of the low affinity binding assay developed by Rexach's group [[53] Cell, 129, 83-96] allowed us to demonstrate that Ssa proteins can mediate interaction between Nup116p and tRNAs in vitro (Figure 7). We now postulate that Ssa proteins may interact with the NPC through specific interaction with certain Nups, such as Nup116p. Indeed, several reports indicate that even importin β has different affinities to different Nups [Ben-Efraim and Gerace (2001), J Cell Biol, 152, 411-417: Phytila and Rexach (2003) J Biol Chem, 278, 42699-42709]. We also mentioned this assertion in Discussion.

*2) tRNA binding. Unfortunately, tRNA in vitro and in vivo binding do not entirely match because both Ssa1 and Ssa2 bind tRNAs in vitro whereas as only Ssa2 does so in vivo. Whereas it is quite possible that factors are missing in vitro that restrict Ssa1 from binding tRNA in vivo, the authors should be cautious. First, the authors show that the label transfer assay allows both Ssa1 and Ssa2 to bind non-tRNA substrates (*Figure 4—figure supplement 1*). Second, only a single tRNA is reported for the in vivo pull-down studies and only 2 tRNAs are monitored in vivo. Regarding tRNA FISH,*
Figure 2
*(SD* ssa1Δ*) shows an affect on tRNAPro nuclear accumulation whereas accumulation is ssa1Δ independent in the statistics,*
Figure 2
*and*
Figure 3*; is there a problem with*
Figure 2*?*

We still do not understand the reason why our tRNA binding data in vitro and FISH data in vivo do not match completely, and we are in our way to investigate this problem for future publication. As commented by the review members, we also suspect that some factor(s) is missing in our in vitro analysis, or that there is labor assignment between Ssa1p and Ssa2p among different isodecoder tRNAs. We had not analyzed every yeast isodecoder in FISH, but additional several FISH analyses yielded similar difference between *ssa1Δ* and *ssa2Δ* so far (see Figure 2—figure supplement 2, a newly added supplement of this figure, and corresponding description in Results). Thus, we still think that Ssa2p is mainly involved in tRNA import while contribution of Ssa1p to this process is minor, if any. However, we tone-downed our assertion in the Discussion.

Also, the reviewers pointed the FISH data of *ssa1Δ* in Figure 2 seems to indicate some effect on tRNA import. We noticed that certain cells showed lower accumulation of tRNAs in the starved nuclei in the *ssa1Δ* cells while others were not. In the beginning, we did suspect that *ssa1Δ* might show weak tRNA import defects. This is why we did quantification of NAIs of individual cells to convince ourselves. However, finally, the quantitative data revealed that there is no statistical difference between the wild type and *ssa1Δ* strains. In addition, we adjusted the signal range of FISH images in the same set of experiments with the same high and low cut-off values. This may imply lower accumulation of tRNA-Pro in the panel in question.

*3) Do Ssa2 and Mtr10 independently affect tRNA subcellular dynamics? The authors conclude that Mtr10 and Ssa2 act via separate tRNA nuclear pathways because* ssa2Δ mtr10Δ *cells are more defective in tRNA nuclear import than either mutant alone. However, this effect is subtle (*Figure 3*) and not entirely convincing. Perhaps the argument could be bolstered if*
Figure 2—figure supplement 1
*were of higher quality as an interpretation of the data is that Ssa2 is important for import upon aa starvation, in contrast to Mtr10 that is thought to affect tRNA nuclear import constitutively. In the absence of additional direct documentation for nutrient-dependent tRNA nuclear import, the authors should be cautious not to conclude that Ssa2 functions in tRNA nuclear import upon aa starvation because Ssa2 could be involved in constitutive import and tRNA nuclear accumulation upon nutrient deprivation could be due to decreased re-export to the cytoplasm*.

We thank the reviewers for the valuable suggestions. Although we still believe that statistical difference among *ssa2Δ*, *mtr10*, and *ssa2Δ mtra10*-double is relevant, according to the reviewers' suggestion, we softened out assertion in the revised version (see Results and Discussion).

Regarding to the Ssa2p involvement in tRNA import under the normal growth conditions, we do not have intention to argue that Ssa2p acts as a tRNA import factor only under starvation conditions, but our original Discussion may mislead the readers in that way. As mentioned in Introduction, tRNA import seems to be constitutive under variety of growth conditions. However, there is no positive data that the capacity of tRNA import is constant under any growth conditions. We wanted to present a possibility that regulation of Ssa2p-driven tRNA import pathway may contribute to alteration of tRNA distribution upon nutrient stress to some extent. Therefore, we changed the Discussion according to the reviewers’ suggestion.

*4) Role of co-chaperones?*
Figure 7
*is not of the same high quality as the FISH studies in the other figures. It is not possible to conclude from the images that the results are different for* sis1-121Δ *and* ydj1Δ *than* zuo1Δ*.*

We are sorry for poor FISH images in these mutant strains. Indeed, we suffered from poor preservation of cellular morphologies in these DnaJ mutant strains. We now somehow improved our procedures to prepare samples for hybridization, and replaced the FIHS images (see new Figure 8 and “Experimental Procedures” section).